

# A Nonparametric Statistical Technique for Combining Global Precipitation Datasets: Development and Hydrological Evaluation over the Iberian Peninsula

Md Abul Ehsan  Bhuiyan,[1] Efthymios. I. Nikolopoulos,[1,2] Emmanouil. N. Anagnostou,[1] Pere Quintana-Seguí,[3] Anaïs Barella-Ortiz,[3,4]

[1] Department of Civil and Environmental Engineering, University of Connecticut, Storrs, CT, USA.

[2] Innovative Technologies Center S.A., Athens, Greece.

[3] Ebro Observatory, Ramon Llull University – CSIC, Roquetes (Tarragona), Spain.

[4] Castilla-La Mancha University, Toledo, Spain.

*Correspondence to*: Emmanouil N. Anagnostou (manos@uconn.edu)

**Abstract.** This study investigates the use of a nonparametric, tree-based model, Quantile Regression Forests (QRF), for combining multiple global precipitation datasets and characterizing the uncertainty of the combined product. We used the Iberian Peninsula as the study area, with a study period spanning eleven years (2000–10). Inputs to the QRF model included three satellite precipitation products, CMORPH, PERSIANN, and 3B42 (V7); an atmospheric reanalysis precipitation and air temperature dataset; satellite-derived near-surface daily soil moisture data; and a terrain elevation dataset. We calibrated the

QRF model for two seasons and two terrain elevation categories and used it to generate rainfall ensembles for these conditions. We then carried an evaluation based on a high-resolution, ground-reference precipitation dataset (SAFRAN) available at 5km/1h resolution and further used generated ensembles to force a distributed hydrological model (the SURFEX land-surface model and the RAPID river routing scheme). To evaluate relative improvements and the overall impact of the combined product in hydrological response, we compared its streamflow simulation results with the results of simulations

from the individual global precipitation and reference datasets. We concluded that the proposed technique could generate realizations that successfully encapsulate the reference precipitation and provide significant improvement in streamflow simulations, with reduction in systematic and random error on the order of 20%–99% and 44%–88%, respectively, when considering the ensemble mean.

**Keywords:** satellite, precipitation, reanalysis, error modeling.



## 1. Introduction

Accurate estimates of precipitation at the global scale, which are essential to hydrometeorological applications (Stephens and Kummerow, 2007), rely primarily on satellite-based observations and atmospheric reanalysis simulations. Although advancement in both satellite retrievals and reanalysis-based precipitation datasets has been continuous (Seyyedi et. al.,

2014; Dee et al., 2011; Huffman et al., 2007; Mo et al., 2012), they are still associated with several sources of error (Derin et al., 2016; Mei et al., 2014; Seyyedi et. al., 2014; Gottschalck et al., 2005; Peña-Arancibia et al., 2013) that limit their use in water resource applications. Quantifying and correcting the sources of the error and characterizing its propagation are important to improving and promoting the use of satellite and reanalysis precipitation estimates in hydrological applications at global scale.

During the past two decades, research investigations have focused on characterizing the error in satellite precipitation products and its propagation in streamflow simulations (Hossain and Anagnostou, 2004; Li et al., 2009; Bitew and Gebremichael, 2011; Nikolopoulos et al., 2013; Mei et al., 2016). These studies have highlighted the dependence of the error on a multitude of factors, including seasonality, topography, soil wetness, and vegetation cover (Derin et al., 2016; Mei et al., 2014; Seyyedi et al., 2014; Hou et al., 2014). Other studies have used stochastic satellite-rainfall error models to

investigate uncertainty characteristics and their dependencies (Hossain and Anagnostou, 2006; Teo and Grimes, 2007; Maggioni et al., 2014; Adler et al., 2001; AghaKouchak et al., 2009) and have used stochastically generated ensemble rainfall fields as input to hydrological models to study the satellite precipitation uncertainty propagation in the simulation of various hydrological variables. The two-dimensional satellite rainfall error model, for example (SREM2D) (Hossain et al., 2006), was used to evaluate the significance of surface soil moisture (Seyyedi et al., 2014) and seasonality (Maggioni et al.,

2017) in modeling the error structure of satellite rainfall products.

Given the multidimensionality of error dependence and the lack of a clear winner among the various precipitation datasets established by these studies, we argue that, to mitigate the error characteristics, one should combine the different precipitation datasets, taking into account the different climatological and land surface factors. A promising approach to modeling appears to be the application of statistical nonparametric techniques, which allow for efficient combining of

25 information on several factors (Ciach et al., 2007; Gebremichael et al., 2011). In fact, although nonparametric statistical techniques are not widely used in rainfall estimation, some notable examples exist in the literature, with encouraging results. Ciach et al. (2007) established a nonparametric estimation technique based on weather radar data to characterize the uncertainties in radar precipitation estimates as a function of range, temporal scale, and season. Lakhankar et al. (2009) introduced a nonparametric technique to retrieve soil moisture from satellite remote sensing products in reliable ways with

30 sufficient accuracy. Moreover, Gebremichael et al. (2011) developed a nonparametric technique for the satellite rainfall error modeling using rain gauge–adjusted, ground-based radar rainfall and reported improved satellite precipitation performance with relatively large variations at low and high rainfall rates.





The use of nonparametric statistical techniques in error modeling has also gained popularity in weather forecasting, climate change prediction, and the modeling of hydrological processes (Croley et al., 2012; Brown et al., 2010; Mujumdar et al., 2008; Yenigun et al., 2013). Recently, a wide variety of nonparametric techniques have been developed for error analytics (Taillardat et al., 2016; He et al., 2015). Nonparametric statistical techniques require fewer assumptions for the form of the

relationship and data. The advantages over parametric techniques for prediction are explained in detail in Guikema et al. (2010). Specifically, the authors exhibited better results (lower prediction error) with nonparametric techniques than with parametric analysis models. The techniques they used were Classification and Regression Trees (CART) and Bayesian Additive Regression Trees (BART) (Chipman et al., 2010). Another nonparametric technique, Random Forest (RF) regression, which provides information about the full conditional distribution of the response variable was used by Breiman,

(2001) and found to yield more robust predictions by stretching the use of the training data partition.

This paper investigates the use of a nonparametric statistical technique for optimally combining globally available precipitation sources from satellite and reanalysis products. Specifically, we use the quantile regression forests (QRF) tree-based regression model (Meinshausen, 2006) to combine dynamic (for example, temperature and soil moisture) and static (for example, elevation) land surface variables with multiple global precipitation sources to stochastically generate improved

precipitation ensembles. The proposed framework provides a consistent formalism for optimally combining several rainfall products by using information from these datasets. It is, furthermore, able to characterize uncertainty through the ensemble representation of the combined precipitation product. We present the development of the proposed framework and evaluate relative improvements in the combined rainfall product in detail. We also evaluate the new combined product in terms of hydrological simulations to assess the importance of precipitation improvement for streamflow simulations, thus highlighting

the usefulness of this approach for global hydrological applications.

The paper is structured as follows: Section 2 briefly explains the study area and the datasets used. Section 3 describes the QRF model, the rainfall error analysis, and the hydrological model setup. Performance evaluation of the combined product in precipitation and corresponding hydrological simulations is presented in section 4. Conclusions and recommendations are discussed in section 5.

**2. Study Area and Data**

The study area we selected for this investigation is the Iberian Peninsula, which has three main climatic zones: Mediterranean, oceanic, and semiarid. The peninsula's climate is primarily Mediterranean, except in its northern and southern parts, which are characterized mostly as oceanic and semiarid, respectively. The topography varies from almost zero elevation to altitudes of 3,500 meters in the Pyrenees. For the hydrological analysis, we focused the study over the Ebro

River basin and, specifically, on five subbasins of different spatial scale: (1) the Ebro River at Tortosa (84,230 km$^2$); (2) the Ebro River at Zaragoza (40,434 km$^2$); (3) the Cinca River at Fraga (9,612 km$^2$); (4) the Segre River at Lleida (11,369 km$^2$); and (5) the Jalon River at Grisen (9,694 km$^2$) (Figure 1). The datasets we used are described below.





## 2.1. Reference precipitation (SAFRAN)

The default reference dataset was recently created by Quintana-Seguí et al. (2016, 2017) using the SAFRAN meteorological analysis system (Durand et al., 1993), which is the same to the one used in earlier studies over France (Quintana-Seguí et al., 2008; Vidal et al., 2010). SAFRAN uses optimal interpolation to combine the outputs of a meteorological model and all available observations, which in this case were provided by the Spanish State Meteorological Agency (AEMET). The variables analyzed were precipitation, temperature, relative humidity, wind speed, and cloudiness. In the case of precipitation, the first guess was deduced from the observations themselves instead of coming from a numerical model, like the other variables. The observations were analyzed daily (as opposed to every six hours for the other variables), but the resulting product had a time resolution of one hour. This was achieved by an interpolation method that used relative humidity to distribute precipitation throughout the day. Spatially, the outputs are presented on a regular grid of 5 km of resolution. The dataset (Quintana-Seguí et al, 2016), which spans 35 years, covers mainland Spain and the Balearic islands.

## 2.2. Satellite-based precipitation

We used three quasi-global satellite precipitation products—CMORPH, PERSIANN, and 3B42 (V7)—in this study. CMORPH (Climate Prediction Center Morphing technique of the National Oceanic and Atmospheric Administration, or NOAA) is a global precipitation product based on passive microwave (PMW) satellite precipitation fields spatially propagated by motion vectors calculated from infrared (IR) data (Joyce et al., 2004). PERSIANN (Precipitation Estimation from Remotely Sensed Information using Artificial Neural Networks) is IR based, and uses a neutral network technique to connect IR observations to PMW rainfall estimates (Sorooshian et al., 2000). TMPA (Tropical Rainfall Measuring Mission Multisatellite Precipitation Analysis), or 3B42 (V7), is a merged IR and passive microwave precipitation product from NASA that is gauge-adjusted and available in both near-real time and post–real time (Huffman et al., 2010). Spatial and temporal resolutions of the satellite precipitation products are $0.25^0$ and three-hourly time intervals, respectively.

## 2.3. Atmospheric reanalysis

For meteorological forcing, we selected the WATCH1 (Water and Global Change FP7 project) Forcing Dataset ERA-Interim (hereafter WFDEI) (Weedon et al., 2014), a contemporary state-of-the-art database. WFDEI, a dataset that follows up on the European Union's WATCH project (Harding et al., 2011), is built on the ECMWF ERA–Interim reanalysis (Dee et al., 2011) with a geographical resolution of $0.5° \times 0.5°$ and a sequential frequency of three hours for the time span 1979–2012, with particular bias corrections using gridded monitoring. Finally, we chose two atmospheric products (atmospheric precipitation and air temperature) among WFDEI variables as predictors for the nonparametric statistical technique.



## 2.4. Soil moisture

The soil moisture information used in this study was obtained from the satellite-based soil moisture estimates produced by the European Space Agency (ESA) Climate Change Initiative (CCI) project under the ESA Programme on Global Monitoring of Essential Climate Variables (ECV) (Liu et al., 2011; Owe et al., 2008; De Jeu, 2003; http://www.esa-

5 soilmoisture-cci.org/node/145). The ESA CCI soil moisture product is derived from passive and active microwave satellite-based sensors (Liu et al., 2011; Liu et al., 2012; Wagner et al., 2012) and provides information on daily surface soil moisture at 0.25o spatial resolution and quasi-global scale.

## 2.5. Terrain elevation

The Shuttle Radar Topography Mission (SRTM) dataset included in this study has, in recent years, been one of the most

10 extensively used publicly accessible terrain elevation datasets. Available at ~90 m spatial resolution, it was obtained using 1-degree digital elevation model (DEM) tiles from the U.S. Geological Survey and interpolated to the 0.25° grid resolution to match the resolution of precipitation and soil moisture products.

## 3. Methodology

### 3.1 Blending technique

In this study, we applied a nonparametric, tree-based regression model, the quantile regression forests (QRF) (Meinshausen, 2006), to produce rainfall ensembles with respect to the reference precipitation. The model input includes the three global satellite precipitation datasets (CMORPH, PERSIANN, and 3B42-V7), the global reanalysis rainfall and air temperature datasets, and the satellite near-surface soil moisture and terrain elevation datasets, described in the previous section. All data were mapped to the grid resolution of 0.25° chosen to be the final spatial resolution for the combined product.

QRF is derived from random forest (Meinshausen, 2006), which is capable of handling data from large samples; it has desirable built-in features, such as variable selection, interaction detection, incorporation of missing data, and the ability to save the trained model for future prediction (Nateghi et al., 2014). QRF provides a nonparametric way to evaluate conditional quantiles for high-dimensional predictors of variables. The conditional distribution function of Y is defined by

$$\hat{F}(y|X=x) = P(Y \leq y \,|X=x) = E\big(1_{\{Y \leq y\}}\,|X=x\big) \qquad (1)$$

where $Y$ is observations of the response variable, $X$ is a covariate or predictor variable, and $E$ is the conditional mean. $E(1_{\{Y \leq y\}}\,|X=x)$, which is approximated by the weighted mean over the observation of $1_{\{Y \leq y\}}$ (Meinshausen, 2006).Then equation (1) can be expressed as

$$\hat{F}(y|X=x) = \sum_{i=1}^{n} \omega_i\,(x) 1_{\{Y_i \leq y\}} \qquad (2)$$



where weight vector $\omega_i(x) = k^{-1} \sum_{i=1}^{k} \omega_i(x, \theta_t)$ using random forests; $k$ indicates number of single trees; each tree built with an i.i.d. (independent and identically distributed) vector $\theta_t$, tree, $t = 1, \ldots, k$ (Meinshausen, 2006).

This nonparametric technique utilizes the weighted average of all trees to compute the empirical distribution function. It keeps not only the mean but also all observation values in nodes and, building on this information, it calculates the conditional distribution. In this method, consistency of the empirical quantities is induced based on a large number of instances in terminal nodes. The overall framework of the QRF scheme is shown in Figure 2. In the error model, we used a random forest of 1,000 trees for each terminal node and created the empirical distribution for each grid cell. For each grid cell, 95% prediction intervals were calculated, which covered the new observation of the response variable with high probability.

To build the rainfall error model, we grouped available rainfall estimates from all the products (three satellite and reanalysis) into three subsets: (1) all rainfall products that report rainfall greater than zero; (2) all rainfall products that report zero rainfall; and (3) at least one product that reports nonzero rainfall. We categorized each case into two seasons: the "warm season," which included data from May through October, and the "cold season," which included data from November through April. We then classified each season category into two levels based on two terrain elevation ranges: above (high) and below (low) 1,000 m.a.s.l. Finally, for each subset, we prepared four groups (warm-high, warm-low, cold-high, cold-low) for the error model. Within each subset, we tested each group by training the rest of the groups. When one of the groups was indicated as a validation dataset, we combined the other three into a training dataset. Later, we estimated the combined optimal precipitation products for each grid cell for all three subsets.

## 3.2. Hydrological modeling

To perform the streamflow simulations, we used the SASER (SAfran-Surfex-Eaudyssée-Rapid) hydrological modeling suite. SASER is a physically based and distributed hydrological model for Spain based on SURFEX (Surface Externalisée), a land-surface modeling platform developed by Météo-France (Masson et al., 2013) that integrates several schemes for different kinds of surfaces (natural, urban, lakes, and so on). The scheme for natural surfaces, ISBA (Noilhan and Planton, 1989; Noilhan and Mahfouf, 1996), has different versions, with differing degrees of complexity. Within SASER we used the explicit multilayer version (Boone, 2000; Decharme et al., 2011) with prescribed vegetation. The physiography was provided by the ECOCLIMAP dataset (Champeaux et al., 2005).

Since SURFEX has no river routing scheme, we chose the RAPID river routing scheme (David et al., 2011a, 2011b) within the Eau-dyssée framework (http://www.geosciences.mines-paristech.fr/fr/equipes/systemes-hydrologiques-et-reservoirs/projets/eau-dyssee). Eau-dyssée transfers SURFEX runoff (surface and subsurface or drainage) from the SURFEX grid cells to the river cells using its own isochrony algorithm. Then, RAPID uses a matrix-based version of the Muskingum method to calculate flow and volume of water for each reach of a river network. The current application of SASER uses HYDROSHEDS (Lehner et al., 2008) to describe the river network. As the current setup cannot simulate dams, canals, or





irrigation, the resulting riverflows are estimations of the natural system (that is, the system without direct human intervention in the form of irrigation or hydraulic infrastructure, such as dams or canals).

It is important to note that, since the current version of SASER uses the default parameters for its different schemes, it has not been specifically calibrated for the target basin. This has some implications. The benefit is that the model is not over

fitted, which makes it directly comparable to global applications of SURFEX, which are not calibrated, either. The downside is that the model might not perform optimally. In the future, we plan to improve the options used in the LSM to adapt its structure better to the necessary physical processes that take place in the basin, while limiting the need for parameter calibration. For the purpose of this study, which involved the relative comparison of multiple rainfall forcing–based simulations, the current model setup was considered adequate.

**3.2. Metrics of model performance evaluation**

We based quantification of the systematic and random error of model-generated ensemble members on different error metrics. We evaluated the random error component based on the normalized centered root mean square error (NCRMSE), which is defined as

$$NCRMSE = \frac{\sqrt{\frac{1}{n}\sum_{i=1}^{n}\left[\hat{y}_i - y_i - \frac{1}{n}\sum_{i=1}^{n}(\hat{y}_i - y_i)\right]^2}}{\frac{1}{n}\sum_{i=1}^{n}y_i} \tag{3}$$

Note that $y_i$ is reference rainfall and $\hat{y}_i$ is estimated rainfall from the blended technique. A NCRMSE value of 0 indicates no

random error, while 1 indicates random error is equal to 100% of the mean reference rainfall.

To measure the systematic error, we used the bias ratio (BR) metric, which indicates the mean of the ratio of estimated rainfall to reference rainfall and is defined as

$$BR = \frac{1}{n}\sum_{i=1}^{n}\left(\frac{\hat{y}_i}{y_i}\right) \tag{4}$$

For an unbiased model, the BR would be 1.

To assess the ability of QRF-generated ensembles to encapsulate the reference rainfall, we used the exceedance probability (EP) metric, which indicates the probability that the reference value will exceed the prediction interval:

$$EP = 1 - \frac{1}{n}\sum_{i=1}^{n}1_{\{Q_{lower_i}<y_i<Q_{upper_i}\}} \tag{5}$$

Here, $Q_{lower}$ and $Q_{upper}$ denote lower and upper boundaries of prediction interval, respectively. The EP would be 0 for an ideal model; that means a perfect encapsulation of the reference within the prediction interval.

To evaluate the accuracy of the QRF-generated ensembles, we used the uncertainty ratio (UR), which measures uncertainty

from the prediction interval ($Q_{lower}$, $Q_{upper}$), as used in (3):





$$UR = \frac{\sum_{i=1}^{n}(Q_{upper} - Q_{lower})}{\sum_{i=1}^{n} y_i} \qquad (6)$$

To achieve accurate and successful prediction, comparatively small prediction intervals are expected. The UR value 1 means best estimate of the actual uncertainty, while a value greater or less than 1 is considered less than perfect.

For the evaluation of the accuracy of the ensembles, we also calculated rank histogram, which is computed by counting the rank of observations, comparing with values from a compiled ensemble, in ascending order. Rank of actual value is denoted by $r_j$ (r1, r2, . . ., $r_{m+1}$) and $r_j$ is expressed as follows:

$$r_j = \hat{P}\{\hat{y}_{i,j-1} < y_i < \hat{y}_{i,j}\} \qquad (7)$$

A flat rank histogram diagram means precise prediction of error distribution (Hamil, 2001; Hamil and Colucci, 1997). A U-shape rank histogram (convex) represents conditional biases, and a concave shape means over-spread. Skewed to the right denotes negative bias and vice versa.

Nash-Sutcliffe Efficiency (NSE) is widely used in hydrology to assess model performance (Nash and Sutcliffe, 1970) and is defined as

$$NSE = 1 - \frac{\sum_{i=1}^{n}(\hat{y}_i - y_i)^2}{\sum_{i=1}^{n}(y_i - \bar{y})^2} \qquad (8)$$

NSE bounds from negative infinity to positive 1, where positive 1 means ideal consistency. A negative value of NSE denotes worse performance of the estimator than the mean of reference.

## 4. Results and discussion

### 4.1 Evaluation of the blending technique

The time series of 20 ensemble-members cumulative rainfall simulated by QRF for high and low elevations in both seasons are shown in Figure 3. The ensemble envelope encapsulates the actual rainfall time series, with better convergence of QRF ensemble members for the warm season but with overall satisfying results for the cold season as well. As is shown, the model was capable of generating stochastic realizations that successfully encapsulated the reference precipitation dynamics. Apart from the ensemble performance, one can note from the results in Figure 3 the variability in the performance of different precipitation products and their inconsistencies relative to the reference precipitation. For example, PERSIANN overestimated for high elevation in the warm season, while CMORPH, 3B42 (V7) and atmospheric reanalysis precipitation underestimated. Overall, CMORPH underestimated the most for both seasons, which indicated poor performance for the QRF ensemble.

Figure 4 compares the combined rainfall product precipitation accumulation maps to the corresponding reference rainfall accumulation maps for the warm and cold seasons. In general, the spatial distribution of rainfall was consistent for both seasons, with the northwestern part of the study area, which is near to the ocean, associated with more precipitation. In the



cold season, the combined product gave higher precipitation in the southwestern and northwestern parts of the Iberian Peninsula than in the warm season. These patterns were consistent with those presented in the reference dataset.

We calculated NCRMSE for the various precipitation products (Figure 5) for five reference precipitation categories, with values in the percentile ranges of <25th, 25th–50th, 50th–75th, 75th–95th, and >95th. The results showed the QRF-based combined product could reduce the random error for all rainfall rate categories in both seasons. It is also shown that the random error reduced consistently in all products as the rainfall rate increased.

To quantify the performance of the combined product in contrast to the individual precipitation datasets, we calculated the relative reduction of the NCRMSE for the different precipitation ranges; Figure 5 presents these performance metric statistics. The relative reduction of the values was defined as the difference between the average of the different datasets and the combined product over the average NCRMSE of the datasets. We noted that relative NCRMSE reduction was greater during the cold season, particularly in regions of low elevation (75% to 99%). During the warm season, the relative reduction varied between 53% and 81% for both high and low elevations. Overall, results from all metrics examined showed that the random error of the combined product was significantly lower than those of the individual global precipitation datasets used in the technique.

The combined product's accuracy was further assessed using BR for both seasons (Figure 6). The results indicated QRF improved accuracy for rain rates beyond the 50th-percentile threshold, exhibiting lower BR values. For moderate to high rainfall in both seasons, all individual rainfall datasets exhibited underestimation, which was reduced in the combined product. In terms of elevation, the magnitude of BR was considerably less for the high elevations in the warm season. The model used elevation as a control parameter, which reflected its ability to reduce noticeably the systematic error at high elevations. For the higher rain rate category (>95th percentile), the BR value in the warm season ranged between 0.5 and 0.6, which was close to estimations for the cold season over the study area. The relative reduction was high for the combined product, varied from 17%−76% for both seasons. Overall, the combined product exhibited BR values closer to 1 than did the individual precipitation datasets, demonstrating superior performance.

Results for exceedance probability (EP) values (Figure 7), which are used to assess the ability of the QRF-generated ensembles to encapsulate the reference data suggest that reference precipitation is well captured within the ensemble envelope. Specifically, considerably reduced exceedance probability values (<0.26) were reported for rain rates below the 95th-percentile threshold for both seasons. A season-based comparison revealed that cold-season EP values were smaller than the corresponding values for the warm season across all rain rate thresholds. Even for the high rain rates (>95th percentile), EP values were found to be acceptable (~0.5).

Analysis of UR showed the QRF-ensemble envelope was associated with slightly wider prediction intervals in the warm season than in the cold, indicating varying degrees of uncertainty throughout the year (Figure 7). A UR closer to 1 was exhibited for higher rain rates, demonstrating that the ensemble envelope represented well the uncertainty for the moderate to high rain rates, while the variability of the ensemble envelope for rain rates below the 50th-percentile threshold overestimated the product uncertainty.



Finally, to evaluate the accuracy of the QRF ensembles, we calculated the rank histogram for the cold and warm seasons. Figure 8 shows the rank histogram of reference values in the posterior sample of QRF ensemble predictions for both seasons. QRF produced a nearly uniformly distributed rank histogram for low to moderate rain rates, which means the rank test was more promising for QRF ensemble predictions for those rain rates. However, the rank histogram exhibited larger values on the right hand side, indicating underestimation of high rain rates in the ensemble prediction, which is potentially attributed to the inclusion in the training dataset of the entire spectrum of values (i.e. large sample of zeroes and low values) that although allow for reproducing certain important features of precipitation (e.g. intermittency) impact the simulation of high rainfall regime.

## 4.2 Evaluation of ensemble hydrological simulations

This section summarizes the results of the streamflow simulations associated with the different precipitation forcing data (satellite, reanalysis, and combined product). We carried out the evaluation using the SAFRAN-based simulations for reference. Comparisons allowed us to understand the performance of each individual precipitation product in terms of streamflow simulations and to evaluate the impact of QRF-based blending technique in terms of hydrological simulations. The NCRMSE and BR quantitative statistics are used here to assess the performance of basin-scale precipitation forcing data and corresponding generated streamflows.

Since SASER does not simulate dams or canals or irrigation, the simulated streamflow reflects how the flow would be if the water resources of the basin were not managed (that is, naturalized streamflow). The Ebro basin is heavily managed, with hundreds of dams and an important canal network. This raised the problem that model flows could not be compared to the observed flows on those subbasins that are influenced by water management, which is most of them. The Spanish Ministry of Agriculture, Fisheries, Food, and the Environment (MAPAMA) had, however, produced monthly naturalized riverflows using the SIMPA rain-runoff model (Ruiz et al., 1998, Álvarez et al., 2004). These are the reference values used by the managers, and they currently offer the only means of reference for validating SASER results.

Table 1 compares the bias and NSE of SASER to those of SIMPA. In terms of monthly accumulation of precipitation, the precipitation data used by SIMPA were similar to SAFRAN's; thus, the difference may have resided in evapotranspiration, which is calculated differently in both models, in terms both of formulation and land-use maps. In terms of NSE, the scores are acceptable at the outlet (between 0.4 and 0.6) and better at most Pyrenean basins (between 0.4 and 0.8), with some exceptions.

Although SASER had room for improvement, mainly with regard to bias, the model was generally able to simulate the main dynamics of the basin. In fact, given that it is essentially evaluated against another model (SIMPA) the reported bias cannot be attributed with certainty to a deficiency of SASER model and therefore this evaluation exercise mostly represents that the model we used can represents flows consistently (to a certain degree) with another model that is widely used in the region. As this study aimed to evaluate streamflow simulations in a relative sense (that is, with respect to SAFRAN-based simulations) and not to reproduce the observed riverflows with precision, the current version of the SASER model was




considered adequate for this purpose. Being physically based, it simulated realistically the interplay among different physical processes, and, thus, it could be used to assess their impact on uncertainty.

To provide an overview of the differences in error magnitude between forcing and response variables, we present our analysis of error metrics for simulated streamflow along with corresponding error metrics in basin-average precipitation.

Results for NCRMSE are shown in Figure 9, which notes that for the two larger basins, Ebro at Tortosa (84,230 km2) and Ebro at Zaragoza (40,434 km2), the NCRMSE (0.1–0.3) of the combined product was significantly lower than those of the other subbasins for all intervals of streamflow values; this was related to the significant smoothing effect on random error associated with larger basin scales.

Consistent with streamflow, these two basins also exhibited considerably lower NCRMSE values (0.2–0.5) for the combined

product in terms of basin-average precipitation. The random component of error was generally slightly higher for low precipitation rates than for the moderate and higher rates for the five subbasins. Similar findings in NCRMSE values for the low streamflow rates were observed. For the basin-average precipitation, the relative NCRMSE reduction was high for the combined product, ranging from 84% to 88% (below the 25th-percentile group). A product-wise comparison showed the combined product had more significant error-dampening effects than reanalysis and satellite precipitation products in

streamflow simulation for all the subbasins. Specifically, the combined product (above the 50th percentile) was characterized by noticeably relative reduction error (44% to 78%) for streamflow. Moreover, we also observed that relative error reduction for the combined product decreased remarkably (56% to 88%), for low streamflow. These results indicated random error was reduced through the rainfall–streamflow transformation in all subbasins. Overall, results show combining information from reanalysis and satellite precipitation datasets could decrease random error in streamflow simulations.

Bias ratio (BR) for basin-average precipitation ranges from overestimation to underestimation as a function of precipitation magnitude, with precipitation rates above the 50th percentile strongly underestimated (BR in the range of 0.07–0.25) (Figure 10). The magnitude of BR for precipitation indicated lower systematic errors in estimates of low to high basin-average precipitation for all subbasins. The corresponding BR values for the simulated streamflows provided a general appreciation of how the magnitude of systematic error in basin-average precipitation translates to systematic error in streamflow

simulations. While a one-to-one correspondence between rainfall and streamflow classes was not possible (note that an event from the highest rainfall class might have resulted in moderate flow values depending on antecedent conditions and so on), we could, however, compare the overall range of BR values between basin-average precipitation and streamflow. As Figure10 indicates for the combined product, BR values were closer to 1 for the different streamflow classes, indicating streamflow was relatively stable. For the large two basins, Ebro at Tortosa and Ebro at Zaragoza, the combined product

underestimated actual values slightly. Overall, the relative systematic error reduction for stream flow ranged from 20% to 99%. These results highlight the usefulness of optimally combining satellite and reanalysis precipitation datasets. Overall, after reducing systematic error, the QRF-generated ensemble corrections brought rainfall products closer to the reference rainfall and simulated runoff.





## 5. Conclusions

A new framework was presented in this study that uses a nonparametric technique (QRF) to combine multiple globally available data sources, including reanalysis and gauge-adjusted satellite precipitation datasets, for generating an improved ensemble precipitation product. The study investigated the accuracy of the combined product using a high-resolution
reference rainfall dataset (SAFRAN) over the Iberian Peninsula. The QRF-generated ensembles are evaluated in terms of precipitation for both warm- and cold-season weather patterns representing a wide variety of precipitation events. Furthermore, the QRF-based streamflow simulations from a distributed hydrological model are evaluated against the SAFRAN-based simulations for a range of basin scales of the Ebro river basin.

Results from the analysis carried out demonstrate clearly that the proposed blending technique has the potential to generate
realistic precipitation ensembles that are statistically consistent with the reference precipitation and are associated with considerably reduced errors. In terms of seasonality effects, the random error significantly decreased for the combined product with increasing rainfall magnitude and this reduction was greater during the cold season. The systematic error of the combined product varied from over- to underestimation as rain rate increased during both seasons. In terms of elevation, among all individual products, the magnitude of systematic error for the combined product was noticeably decreased for the
higher elevations, which is a strong indication for using the proposed scheme in retrieving global precipitation in high-elevation regions. Overall, the reduction of the combined product (relative to other products) for the systematic and random error ranged between 17%−76% and 53%−99%, respectively.

Evaluation of the impact on streamflow simulations showed that the magnitude of systematic and random error for simulations corresponding to the combined product was significantly lower than for the individual precipitation products. In
addition, for the combined product, the large-scale basins exhibited considerably lower systematic and random error values than the small-size basins, which shows the dependency on basin scale. Specifically, the relative reduction for the combined product in systematic and random error ranged between 20%−99% and 44%−88%, respectively, which highlights the potential of the proposed technique in advancing hydrologic simulations.

Our overall conclusion is that the proposed framework offers a robust way of blending globally available precipitation
datasets providing at the same time an improved precipitation product and characterization of its uncertainty. This can have important applications in studies dealing with water resources reanalysis and quantification of uncertainty in hydrologic simulations. Future work will include evaluation of the proposed framework in different hydroclimatic regions considering also sensitivity of its performance to availability (e.g. record length and spatial coverage) of in situ reference precipitation.

Acknowledgement: This research was supported by the FP7 project eartH2Observe (grant agreement no. 603608).





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







Figure 1. Map of Iberian Peninsula case study area.


Figure 2: General framework of the Quantile Regression Forests (QRF) scheme.





Figure 3: Time series of cumulative rainfall of CMORPH, PERSIANN, 3B42 (V7), re-analysis rainfall, and QRF ensemble
5   members (blue envelop) for warm and cold season.



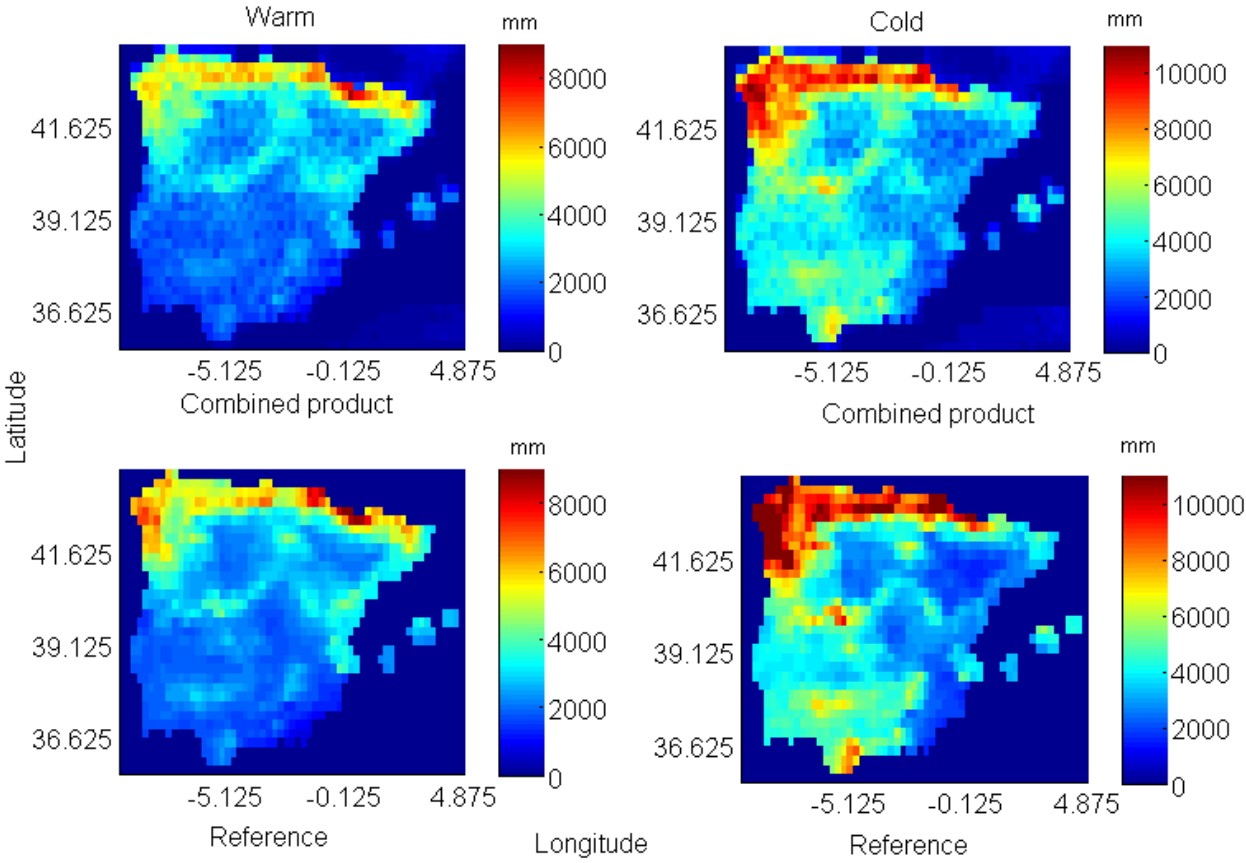

Figure 4: QRF-generated mean ensemble and reference rainfall maps for warm and cold season.





Figure 5: Normalized Centered Root Mean Square error for warm and cold season.





Figure6: Bias ratio for warm and cold season.





Figure 7: Exceedance probability and Uncertainty ratio for warm and cold season





Figure 8: Rank histogram for warm and cold season





Figure 9: Normalized Centered Root Mean Square error for basin-average rainfall and streamflow.





Figure 10: Bias ratio of precipitation and streamflow.




Table1: The bias and NSE of high resolution SAFRAN-SURFEX simulation

| River | Station | Area(km2) | NSE | Bias |
|---|---|---|---|---|
| Ega | Andosilla | 1445 | 0.735 | -13.52 |
| Arga | Funes | 2759 | 0.651 | -38.068 |
| Aragón | Caparroso | 5462 | 0.733 | -25.655 |
| Jiloca | Daroca | 2202 | 0.069 | -23.28 |
| Ebro | Zaragoza | 40434 | 0.716 | -29.324 |
| Gállego | Ardisa | 2040 | 0.477 | -26.613 |
| Ésera | Graus | 893 | 0.37 | -28.177 |
| Cinca | El Grado | 2127 | 0.629 | -21.223 |
| Segre | Lleida | 11369 | 0.256 | -35.016 |
| Ebro | Tortosa | 84230 | 0.576 | -27.2 |
| Najerilla | Mansilla | 242 | 0.114 | -62.214 |
| Albercos | Ortigosa | 45 | 0.193 | -45.826 |
| Cidacos | Yanguas | 223 | 0.392 | -42.551 |
| Salazar | Aspurz | 396 | 0.731 | -27.012 |
| Irati | Liedena | 1546 | 0.721 | -30.451 |
| Arga | Echauri | 1756 | 0.508 | -46.398 |
| Ega | Estella | 943 | 0.622 | -36.172 |
| Aragón | Yesa | 2191 | 0.716 | -22.334 |
| Noguera Pallaresa | Collegats | 1518 | -0.768 | 2.102 |
| Huerva | Mezcalocha | 620 | 0.427 | -24.375 |
| Ebro | Mendavia | 12010 | 0.592 | -36.507 |
| Ebro | Flix | 82416 | 0.59 | -27.564 |
| Ésera | Barasona | 1511 | 0.446 | -32.846 |
| Ebro | Ribarroja | 81060 | 0.595 | -27.604 |
| Jalón | Calatayud | 6841 | 0.222 | -22.072 |

