# Peer review of "A Nonparametric Statistical Technique for Combining Global Precipitation Datasets: Development and Hydrological Evaluation over the Iberian Peninsula"

_Hydrology and Earth System Sciences, 2017_

## Short Comment (SC1) · 9 Jun 2017

**Short Comments**

I quickly read the paper by Md Abul Ehsan Bhuiyan et al. as I am very interested to the proposed methodology. Indeed, as the authors might know, we are working on the combination of state-of-the-art precipitation products (e.g., CMORPH, PERSIANN, 3B42) and satellite soil moisture data (e.g., ESA CCI SM) for improving satellite rainfall

estimate (over land). I believe the paper is well written and clear. The final results are very encouraging. However, in my opinion a better description of the different steps involved in the procedure should be given. I reported below my comments/suggestions that I guess could be used from the authors for improving the paper's relevance.

1) As mentioned above, I am very interested to understand the contribution of the different datasets to the final combined precipitation dataset. What is the contribution of the satellite products with respect to the reanalysis? Which is the contribution of satellite soil moisture data? And of air temperature? I believe that running the QRF model in different scenarios considering different subsets of data will easily allow to reply to these questions.

2) Actually, if I well understood, the same data period is used for the calibration and the assessment of the combined precipitation dataset. It is not fair in the comparison with the single products. Likely, a split of the data in a calibration/validation period is needed.

3) What is the final objective of the paper? If the authors want to provide a superior rainfall dataset, it should be tested against the SAFRAN reference dataset. What are the differences in the performance of hydrological modelling between SAFRAN and the combined dataset? This analysis might provide interesting insights.

4) (MINOR) Among the different satellite rainfall products, PERSIANN and CMORPH should be the versions only based on satellite data. Differently, 3B42 (V7) is corrected with rain gauge observations. Therefore, the comparison between them is not fair, and I suggest in using the real-time version of TMPA (3B42RT) for a more interesting comparison.

---

## Referee Comment (RC1) · V. Maggioni (Referee) · 27 Jun 2017

This study proposes to use a non-parametric statistical model (the Quantile Regression Forest) to merge several precipitation datasets together with ancillary information (e.g., soil moisture, air temperature, and terrain elevation), which are used as predictors to estimate a superior precipitation product. I find this work innovative and worth publication. I have a few comments, which I would like the authors to consider and which I hope will meliorate the manuscript.

[Figure]

Minor comments:

Sometime in the text, there is confusion in the way the words "ensemble" and "members" are used. An ensemble is made of several members. Therefore, "ensembles" would refer to multiple ensembles made of several members.

Page 1, Line 12: replace "generated ensembles to force" with "to generate ensembles that force"

Page 2, Line 18: rephrase as "Satellite rainfall error models, such as SREM2D (Hossain et al., 2006), have been used to..."

Page 2, Line 22: replace "the error characteristics" with "errors and uncertainties"

Page 2, Line 24: replace "allow for efficient combining of" with "efficiently combine"

Figure 2: a "d" is missing in the word "and" in the top center block. This methodological framework scheme could be improved: as it is, it is unclear what each block does.

Figure 6: can the authors discuss why QRF performs so poorly for lower rain percentiles (<25%) in term of bias?

Major comments:

A better explanation on why the authors picked those predictors is needed. For instance, why including soil moisture, but not a vegetation indicator? Why including three satellite precipitation products instead of two? Was the dataset combination that produced the best results picked? Are all the predictors really needed?

What is the impact of merging datasets that are not totally independent? For instance CMORPH and TMPA 3B42 use the same MW overpasses in their algorithms, as PERSIANN, CMORPH, and TMPA use the same IR observations. I am wondering whether there is any chance that too much weight is given to this information in the merging model.

---

## Short Comment (SC2) · 28 Jun 2017

**Response to Interactive discussion**

Hydrology and Earth System Sciences (HESS)

Title: A Nonparametric Statistical Technique for Combining Global Precipitation Datasets:
Development and Hydrological Evaluation over the Iberian Peninsula

Md Abul Ehsan Bhuiyan,[1] Efthymios. I. Nikolopoulos,[1,2] Emmanouil. N. Anagnostou,[1] Pere Quintana-Seguí,[3] Anaïs Barella-Ortiz,[3,4]

We would like to thank Dr. Luca Brocca for his insightful discussion and constructive suggestions. Below we provide a point-by-point response to his comments. Dr. Brocca's comments are in red and our responses in black font.

Short Comments

I quickly read the paper by Md Abul Ehsan Bhuiyan et al. as I am very interested to the proposed methodology. Indeed, as the authors might know, we are working on the combination of state-of-the-art precipitation products (e.g., CMORPH, PERSIANN, 3B42) and satellite soil moisture data (e.g., ESA CCI SM) for improving satellite rainfall C1 HESSD Interactive comment Printer-friendly version Discussion paper estimate (over land). I believe the paper is well written and clear. The final results are very encouraging. However, in my opinion a better description of the different steps involved in the procedure should be given. I reported below my comments/suggestions that I guess could be used from the authors for improving the paper's relevance.

1) As mentioned above, I am very interested to understand the contribution of the different datasets to the final combined precipitation dataset. What is the contribution of the satellite products with respect to the reanalysis? Which is the contribution of satellite soil moisture data? And of air temperature? I believe that running the QRF model in different scenarios considering different subsets of data will easily allow replying to these questions.

**Ans:**

We appreciate reader's point about the contribution of the different datasets to the final combined precipitation dataset. We demonstrate the relative contribution of each variable based on the variable importance methodology (Breiman, 2001) that is widely applied in this context. Briefly, the mean squared error (MSE) computed from the original model (i.e. considering all variables) is compared against MSE from a new model that holds all variables the same as the original model except one, the one we want to determine it's relative importance. Comparison of the MSE between original and new model demonstrates essentially the importance of each variable examined. For more details on the method you can refer to Breiman, 2001.

In this document we present results for the variable importance test only for one of the groups included in our methodology (for warm period-high elevation when rainfall is greater than zero for all products). The importance of the predictor variables depends on the magnitude of the percentage increase in mean square error (%IncMSE) of the model. Higher values of %IncMSE indicate higher importance of the predictor variable.

Figure 1 below shows the variable importance of the seven predictors. According to these results (and for the specific group examined) all variables are important but the level of importance varies considerably between the different variables considered. Soil moisture, reanalysis and CMORPH precipitation rank as most important, with elevation being the least important (see Figure 1 for details).

[Figure]

Figure 1: Variable importance plot, where %IncMSE is the percentage increase in mean square error.

We are going to include a complete analysis in the revised paper.

2) Actually, if I well understood, the same data period is used for the calibration and the assessment of the combined precipitation dataset. It is not fair in the comparison with the single products. Likely, a split of the data in a calibration/validation period is needed.

**Ans:**
First, we would like to clarify that the precipitation error statistics are based on hold-one-out validation. Meaning, each point in the statistics was not included in the calibration of the technique. The hydrologic evaluation on the other hand was based on the final (all points) technique

calibration and it is intended to demonstrate the importance of the ensemble representation and propagation in the hydrologic simulations. This aspect will be better clarified in the revised manuscript.

We would also like to point out that objective of this paper is to present a methodology that allows to combine different sources of information on precipitation and other variables (soil moisture, temperature, terrain complexity, etc.) to provide a more accurate representation of precipitation estimation uncertainty, and through this ensemble representation evaluate how this uncertainty propagates in hydrologic simulations. Results in the current manuscript demonstrate that the Quantile Regression Forests (QRF) technique is successful on doing so, while reducing mean error in both precipitation and flow simulations.

The issue of separating data in calibration/validation is more relevant in the case that we would like to evaluate how stable is the blending algorithm for real-time application, where ground-reference is not yet available. This is an important question, but it is not within the scope of our current manuscript. It is definitely though one to consider as a future step in this line of work.

So, to summarize, our current objectives are more in line with the context of retrospective analysis. Given that a vast majority of studies on hydro-meteorological applications of satellite-precipitation products is based on post-real time products, we consider both valid and significant to be able to apply a method like the one proposed for advancing precipitation reanalysis.

3) What is the final objective of the paper? If the authors want to provide a superior rainfall dataset, it should be tested against the SAFRAN reference dataset. What are the differences in the performance of hydrological modelling between SAFRAN and the combined dataset? This analysis might provide interesting insights.

**Ans:**
As stated above the aim is mostly to provide a method rather than a "superior dataset". We are presenting a blending technique that leads to an improved characterization of precipitation estimation uncertainty through an optimal combination of precipitation and other datasets, but we do not claim that we have examined exhaustively the combinations of variables or products that can potentially lead to a "superior product". In fact we hope that this work will trigger the interest of the community (as in your case) to investigate these aspects in more detail.

Evaluation metrics for rainfall and streamflow simulations are reported in the manuscript based on SAFRAN and SAFRAN-forced simulations as reference respectively. Please take a look at Figs 5,6,9,10 and corresponding text in manuscript.

4) (MINOR) Among the different satellite rainfall products, PERSIANN and CMORPH should be the versions only based on satellite data. Differently, 3B42 (V7) is corrected with rain gauge observations. Therefore, the comparison between them is not fair, and I suggest in using the real-time version of TMPA (3B42RT) for a more interesting comparison.

**Ans:**

Thank you for this note. In fact, the CMORPH and PERSIANN products considered in this work are also gauge-adjusted. We will clarify this in the revised manuscript.

---

## Referee Comment (RC2) · Anonymous Referee #2 · 4 Jul 2017

This study is focused on developing and evaluating a non-parametric statistical method to generate an ensemble of precipitation estimates to better capture the uncertainty in global precipitation estimates. The inputs to the algorithm include three global satellite-based precipitation estimates, a reanalysis product (rainfall and air temperature), elevation, and remotely-senses soil moisture.

While the manuscript is written very well, and the topic will be of interest to the HESS readership, the following caveats needs to be addressed before publication of the manuscript:

[Figure]

-Major Comments:

1) The authors provide detailed information about the products used in the QRF method and explain the method itself very well. But no information is provided on how the training and validation of the method is performed. How much of the data is used for training? How much used for validation and testing? Please also include the temporal coverage of the data.

2) There is also no information on avoiding overfitting. One of the challenges in data-driven methods is overfitting (i.e. the method is so fine tuned to the training data, and has larger errors when applied to new datasets). I don't see any discussion of this in the paper. For example how did you choose to use 1000 trees in the model? Are there noticeable differences between the performance of the method during training and validation?

3) How are the ensembles generated? No information is provided on how each ensemble member is initialized and generated using the QRF trained on the data.

4) The results provided in section 4 needs to be clarified whether they are based on the data used in training or the data used in validation, or a mixture of both.

5) The low value of NCRMSE for the small basins report in Page 11, Line 6 is a signal of overfitting in the algorithm. This is another indication that overfitting should be analyzed in depth.

6) Page 6, Lines 10-18: Please clarify if different trees are developed for the three groups that you introduce at the beginning of the paragraph. You have introduced four groups at the end (warm-high, warm-low, cold-high and cold-low) but there is no reference to the categorization of products based on their rain detection (group 1-3 in lines 11-12).

-Minor Comments:

1) Why did you choose to use PERSIANN product instead of the newer version

PERSIAN-CCS?

2) In section 2.3, please include details on how you have downscaled the 0.5 degree reanalysis product to 0.25 degree to be consistent with other products.

3) In section 2.4, please include the version number of the ESA-CCI product.

4) Page 8, Line 2: What does actual uncertainty mean? Do you mean uncertainty in the reference product? If so, please explain how a UR=1 will provide the best estimate of the uncertainty in the reference product.

---

## Short Comment (SC3) · 26 Jul 2017

**Response to Interactive discussion**

Hydrology and Earth System Sciences (HESS)

Title: A Nonparametric Statistical Technique for Combining Global Precipitation Datasets: Development and Hydrological Evaluation over the Iberian Peninsula

Md Abul Ehsan Bhuiyan,[1] Efthymios. I. Nikolopoulos,[1,2] Emmanouil. N. Anagnostou,[1] Pere Quintana-Seguí,[3] Anaïs Barella-Ortiz,[3,4]

We would like to thank Dr. V. Maggioni for her insightful discussion and constructive suggestions. Below we provide a point-by-point response to her comments. Dr. Maggioni's comments are in red and our responses in black font.

Sometime in the text, there is confusion in the way the words "ensemble" and "members" are used. An ensemble is made of several members. Therefore, "ensembles" would refer to multiple ensembles made of several members

Page 1, Line 12: replace "generated ensembles to force" with "to generate ensembles that force"

Thank you. It will be corrected in the revised paper.

Page 2, Line 18: rephrase as "Satellite rainfall error models, such as SREM2D (Hossain et al., 2006), have been used to. . ."

Thank you. It will be corrected in the revised paper.

Page 2, Line 22: replace "the error characteristics" with "errors and uncertainties"

Thank you. It will be corrected in the revised paper.

Page 2, Line 24: replace "allow for efficient combining of" with "efficiently combine"

Thank you. It will be corrected in the revised paper.

Figure 2: a "d" is missing in the word "and" in the top center block. This methodological framework scheme could be improved: as it is, it is unclear what each block does.

Thank you. It will be corrected in the revised paper. The methodological framework scheme will be updated by improving content and clarity.

Figure 6: can the authors discuss why QRF performs so poorly for lower rain percentiles (<25%) in term of bias?

**Ans:**
Thank you for the comment. Generally, the QRF model is expected not to capture well very low and extremely high values due to the weakness of the empirical distribution function to model

probabilities close to 0 or 1. Moreover, studies have shown that QRF can perform better in generating one-sided prediction intervals, which is the cases in Juban et al. (2007), Francke et al. (2008) and Zimmermann et al. (2012). Nevertheless, we would like to note that these very low rain rates that the model cannot capture well represent a small fraction of the precipitation accumulation.

Major comments:
A better explanation on why the authors picked those predictors is needed. For instance, why including soil moisture, but not a vegetation indicator? Why including three satellite precipitation products instead of two? Was the dataset combination that produced the best results picked? Are all the predictors really needed?
What is the impact of merging datasets that are not totally independent? For instance CMORPH and TMPA 3B42 use the same MW overpasses in their algorithms, as PERSIANN, CMORPH, and TMPA use the same IR observations. I am wondering whether there is any chance that too much weight is given to this information in the merging model.

**Ans:**

Thank you for asking about the contribution of the different predictors to the final combined precipitation product.

We selected all the predictors in our error analysis based on the variable importance methodology (Breiman, 2001), which indicated the level of influence of variables in the model prediction. We are going to include an explanation for choosing all those predictors in the revised paper.

Soil moisture and vegetation are dependent on each other. If vegetation increases, soil moisture increases. For different vegetation, moisture holding capacity is also different. As we had a quality controlled soil moisture dataset available, it was pretty straight forward to use that data, instead of using any vegetation indicator.

Below, we present results for the variable importance test only for one of the groups included in our methodology (for warm period-high elevation when rainfall is greater than zero for all products). The figure shows the variable importance of the seven predictors. According to these results, soil moisture, reanalysis and three satellite precipitation (CMORPH, PERSIANN and 3B42 (V7) were ranked as most important predictors. This is the reason why we choose three satellite precipitation products instead of two to produce best results.

[Figure]

Figure: Variable importance plot, where %IncMSE is the percentage increase in mean square error. Higher values of %IncMSE indicate higher importance of the predictor variable.

The variable importance test for the all predictors showed that the model results closely sensitive to all those variables. Therefore, we select all the dataset combination to produce blending precipitation product.

From the variable importance test, it is clearly shown that the individual satellite precipitation product has a strong impact in model prediction. So, better prediction is expected after merging all those products although they are not totally independent.

We are going to include a complete analysis in the revised paper.

---

## Short Comment (SC4) · 26 Jul 2017

**Response to Interactive discussion**

Hydrology and Earth System Sciences (HESS)

Title: A Nonparametric Statistical Technique for Combining Global Precipitation Datasets: Development and Hydrological Evaluation over the Iberian Peninsula

Md Abul Ehsan Bhuiyan,[1] Efthymios. I. Nikolopoulos,[1,2] Emmanouil. N. Anagnostou,[1] Pere Quintana-Seguí,[3] Anaïs Barella-Ortiz,[3,4]

We would like to thank Reviewer for his insightful discussion and constructive suggestions. Below we provide a point-by-point response to his/her comments. Reviewer's comments are in red and our responses in black font.

-Major Comments:

1) The authors provide detailed information about the products used in the QRF method and explain the method itself very well. But no information is provided on how the training and validation of the method is performed. How much of the data is used for training? How much used for validation and testing? Please also include the temporal coverage of the data.

Ans:

First, we would like to clarify that the precipitation error statistics are based on hold-one-out validation. That means, each data-pair used in the validation statistics was not included in the training of the non-parametric model. This approach gives better estimates of the model performance because it trains and tests based on the entire data set. The temporal resolution of all precipitation products used in this study is three hours. This aspect will be better clarified in the revised manuscript.

2) There is also no information on avoiding overfitting. One of the challenges in data driven methods is overfitting (i.e. the method is so fine tuned to the training data, and has larger errors when applied to new datasets). I don't see any discussion of this in the paper. For example how did you choose to use 1000 trees in the model? Are there noticeable differences between the performance of the method during training and validation?

Ans:

Thank you for bringing the overfitting aspect in the discussion.

First we would like to note that the results showing in section 4 of the original manuscript are based on a validation dataset of 11 years and found to be prominent results for precipitation estimation as well as stream flow simulation. That means our model is successfully calibrated and is able to predict well the independent data.

Quantile Regression Forests (QRF) uses bagged version of decision trees and obtains a lower test error by variance reduction (Meinshausen, 2006). Higher number of trees reduces the variance of the model. So, increasing the number of trees in the ensemble won't have any impact on the bias of the model. Furthermore, a higher variance reduction can be achieved by decreasing the correlation between trees in the ensemble. Therefore, QRF utilizes the optimal number 'mtry' (size of the random subset of predictors) for split point selection at each node. It will introduce some randomness in to the ensemble to reduce the correlation between trees which helps to avoid overfitting (Meinshausen, 2006). In general, prominent '*mtry*' is obtained by cross validation methods in extending the sample size. The application of the machine learning tools can manipulate the training data in such a way that the actual results expected from the unseen data can be quite different from the evaluated results using the training data set, which is called overfitting. Therefore, in this analysis, we used hold-one-out cross validation method which prevent overfitting by producing reliable results. Applying this validation technique, the model has good skill on both the training dataset and the unseen test data.

To strengthen the validation results, in the revised version, we will present validation using one-year-leave-out cross validation. Namely, for each year of the database hold out for validation, we will be calibrating on the rest of the years (ten years). The performance of the combined product, based on the one-year-leave-out cross validation (presented in Figures 1 and 2 below), is found to be very similar to the results shown in the original manuscript (Figures 5 and 6) determined based on the hold-one-out cross validation.

Both validation approaches demonstrated that our model is able to reduce significantly the systematic and random error and is not overfitting.

As we discussed, higher number of trees would reduce the variance of the model and help to avoid overfitting. Therefore, the size of the forest should be relatively large for the stabilizing effect of many trees. For the Quantile Regression Forests, trees are grown as in the standard random forests algorithm and bagged versions of the training data are used for each of the $k = 1000$ trees to determine the optimal number *mtry* (Meinshausen, 2006). Therefore, to demonstrate the stability of QRF, the default value ($k = 1000$) is chosen throughout all simulations.

[Figure]

Figure1: Bias ratio for warm and cold season.

[Figure]

Figure2: Normalized Centered Root Mean Square error for warm and cold season.

 3) How are the ensembles generated? No information is provided on how each ensemble member is initialized and generated using the QRF trained on the data.

Ans:

This is part of the statistical machine learning model. For QRF model, we initialize a random forest of 1,000 trees for each terminal node of each of the classified dataset. We calculate 95% prediction intervals for each grid. QRF utilizes the same weights to calculate the empirical distribution function. When, X is predictor variable and Y is response variable, QRF utilizes a

weighted average of all trees for the predicted expected response values to calculate the empirical distribution. To conduct the hydrological simulations in this study, we resampled from the empirical distribution function for 20 times per grid cell to obtain "reference"-like rainfall ensemble members. We will clarify this aspect in the revised manuscript.

4) The results provided in section 4 needs to be clarified whether they are based on the data used in training or the data used in validation, or a mixture of both.

Ans:

The results provided in section 4 are based on hold-one-out validation as explained in our response to question 1. Our study period spans eleven years (2000–2010) and we validated all those 11 years belonging in our dataset. Validation results are presented in section 4.

5) The low value of NCRMSE for the small basins report in Page 11, Line 6 is a signal of overfitting in the algorithm. This is another indication that overfitting should be analyzed in depth.

Ans:

It's actually not overfitting here. Generally, Overfitting depends on the inconsistency of training and validation model results. Overfitting refers good performance on the training data, poor generalization to validation data. Generalization indicates how well the concepts learned by a training model apply to new dataset. So, if we produce validation and training model for particular group of dataset and find inconsistency between two results, then we can justify overfitting. As we said in section 4, all the results are based on only validation results, there is no way of knowing whether overfitting or under fitting without comparing training results. So it is not possible to justify overfitting in algorithm to examine from the validation results only. Results for NCRMSE are shown in Figure 9, which are consistent in terms of the reduction of the random error for all the subbasins as well as precipitation and streamflow percentile ranges. This is the indication that how we successfully trained our model instead of overfitting.

6) Page 6, Lines 10-18: Please clarify if different trees are developed for the three groups that you introduce at the beginning of the paragraph. You have introduced four groups at the end (warm-high, warm-low, cold-high and cold-low) but there is no reference to the categorization of products based on their rain detection (group 1-3 in lines 11-12).

Ans:

Yes, we developed different trees for the three groups. If we grow similar kind of trees, every sampling will be equal that affects the model results. As we mentioned, (QRF) uses bagged version (bootstrapped aggregating) of decision trees by randomly sampling from bootstrapped sample which reduces variance and helps to avoid overfitting to improve the stability and accuracy of our proposed machine learning algorithms. That is the whole idea of choosing

ensemble method where trees grow independently because of the combination of bootstrap samples and random drawing of variables.

Actually, we classified available rainfall estimates from all the products (three satellite and reanalysis) into three subsets: (1) all rainfall products that report rainfall greater than zero (2) all rainfall products that report zero rainfall; and (3) at least one product that reports nonzero rainfall. Then, for each subset, we created 4 groups:  warm period-high elevation, warm period-low elevation, cold period-high elevation, cold period-low elevation for the error model. Finally, we prepared total 12 groups from all threes subsets (each one has 4 groups) for the error model.

All these classification we created by our own justification to keep similar types of dataset together. If we keep different kinds of dataset together, our model will not be efficient in accurate prediction due to the lack of uniformity in dataset. Generally, the QRF model is expected not to capture well very low and extremely high values due to the weakness of the empirical distribution function to model probabilities close to 0 or 1. The distribution of proper sample size plays an important role in empirical distribution function. Therefore, very large sample sizes required for low and extremely high values to quantify the rate of convergence to the underlying cumulative distribution function. This is the reason we categorized our dataset from above mentioned procedure.

-Minor Comments:

1) Why did you choose to use PERSIANN product instead of the newer version PERSIAN-CCS?

Ans:

In this study we chose to use gauge adjusted satellite precipitation products: 3B42 (V7), CMORPH and PERSIANN. The gauge-adjusted PERSIAN-CCS is not available over the Iberian Peninsula. Using the PERSIAN-CCS in precipitation error analysis is a good suggestion that we could investigated in a future research.

2) In section 2.3, please include details on how you have downscaled the 0.5 degree reanalysis product to 0.25 degree to be consistent with other products.

Ans:

The dataset was interpolated in space and time using the nearest neighbor interpolation technique for every time steps so as to match the other products. We will add text about this aspect in the revised manuscript.

3) In section 2.4, please include the version number of the ESA-CCI product.

Ans:

The version number of the ESA-CCI product is v02.0 which we will add in in the revised manuscript.

4) Page 8, Line 2: What does actual uncertainty mean? Do you mean uncertainty in the reference product? If so, please explain how a UR=1 will provide the best estimate of the uncertainty in the reference product.

Ans:

Thank you for raising this question. Here, actual uncertainty indicates the maximum possible uncertainty of the prediction interval, which is 1. It is not the uncertainty in reference product. Uncertainty Ratio (UR) quantifies the prediction interval width relative to the magnitude of the predicted variable. UR value close to 1, indicates confidence intervals being in the order of magnitude of the predicted values. We will clarify about this aspect in the revised manuscript.

---

## Author Comment (AC1) · 4 Sep 2017

The comment was uploaded in the form of a supplement:
https://www.hydrol-earth-syst-sci-discuss.net/hess-2017-268/hess-2017-268-AC1-supplement.zip

---

## Author Response (AR1)

**Response to comments from interactive discussion**
Hydrology and Earth System Sciences (HESS)
Title: A Nonparametric Statistical Technique for Combining Global Precipitation Datasets:
Development and Hydrological Evaluation over the Iberian Peninsula

Md Abul Ehsan Bhuiyan, Efthymios. I. Nikolopoulos, Emmanouil. N. Anagnostou, Pere
Quintana-Seguí, Anaïs Barella-Ortiz

We would like to thank the reviewers and Editor for their careful reading of this manuscript and
for the insightful comments and constructive suggestions, which we believe has helped to
improve the quality of this manuscript. Our point-to-point responses follows. The reviewers'
comments are in red and our responses in black. Revised or new text in the revised manuscript
are highlighted with red color.

**Response to Referee Comments #1  (RC1):**

This study proposes to use a non-parametric statistical model (the Quantile Regression
Forest) to merge several precipitation datasets together with ancillary information (e.g., soil
moisture, air temperature, and terrain elevation), which are used as predictors to estimate a
superior precipitation product. I find this work innovative and worth publication.
I have a few comments, which I would like the authors to consider and which I hope will
meliorate the manuscript.

Minor comments:
Sometime in the text, there is confusion in the way the words "ensemble" and "members" are
used. An ensemble is made of several members. Therefore, "ensembles" would refer to multiple
ensembles made of several members

**Ans:**
Thank you for the comment. 'Ensemble members' is replaced by 'Ensemble' in the revised
version of the paper.

Page 1, Line 12: replace "generated ensembles to force" with "to generate ensembles
that force"

**Ans:**
We have revised the sentence in the updated version of the manuscript.

Page 2, Line 18: rephrase as "Satellite rainfall error models, such as SREM2D (Hossain
et al., 2006), have been used to. . ."

**Ans:**
It is modified in the revised version of the manuscript.

Page 2, Line 22: replace "the error characteristics" with "errors and uncertainties"

**Ans:**
It is corrected in the revised version of the manuscript.

Page 2, Line 24: replace "allow for efficient combining of" with "efficiently combine"
It is corrected in the revised version of the manuscript.

Figure 2: a "d" is missing in the word "and" in the top center block. This methodological framework scheme could be improved: as it is, it is unclear what each block does.

**Ans:**
It is corrected in the revised version of the manuscript. The methodological framework scheme is updated by improving block 1, 2 and 3 as shown in figure 2. Section 3.1 is also modified with sufficient text based on the updated general framework of the Quantile Regression Forests (QRF) scheme.

Figure 6: can the authors discuss why QRF performs so poorly for lower rain percentiles (<25%) in term of bias?

**Ans:**
 In section 4.2, we added text to discuss about this, which we hope clarifies this issue. For the low rainfall, the systematic error reduction for the combined product was not prominent resulting comparatively higher BR value. Generally, the QRF model is expected not to capture well the two tails of the distribution (very low and high values) due to the weakness of the empirical distribution function to model probabilities close to 0 or 1. Moreover, studies have shown that QRF can perform better in generating one-sided prediction intervals, which was the case in several past studies such in Juban et al. (2007), Francke et al. (2008) and Zimmermann et al. (2012).

Major comments:
A better explanation on why the authors picked those predictors is needed. For instance, why including soil moisture, but not a vegetation indicator? Why including three satellite precipitation products instead of two? Was the dataset combination that produced the best results picked? Are all the predictors really needed?
What is the impact of merging datasets that are not totally independent? For instance CMORPH and TMPA 3B42 use the same MW overpasses in their algorithms, as PERSIANN, CMORPH, and TMPA use the same IR observations. I am wondering whether there is any chance that too much weight is given to this information in the merging model.

**Ans:**
To address this question, we added a new section in revised manuscript (section 4.1) that explains the contribution of the selected variables in model prediction. The new section discusses the reasons behind the selection of the predictive variables for the blending technique supported

by past research. After selecting these variables, the sensitivity analysis is performed based on the variable importance methodology (Breiman, 2001). The variable importance methodology showed that all selected variables impact significantly the model prediction accuracy and should be included in the model. It is noted though that the selected variables should not be considered as the optimum set of variables to be applied globally. Inclusion of additional information across different hydro-climatic regimes can potentially exhibit further improvement of the performance of the blending algorithm.

**Response to Referee Comments #2 (RC2)**

-Major Comments:
1) The authors provide detailed information about the products used in the QRF method and explain the method itself very well. But no information is provided on how the training and validation of the method is performed. How much of the data is used for training? How much used for validation and testing? Please also include the temporal coverage of the data.

**Ans:**
In section **3.1,** we describe the validation and calibration process with new detailed information. For precipitation error statistics, leave-one-pixel-out and leave-one-year-out cross validation are applied. For leave-one-pixel-out, each data-pair used in the validation statistics was treated independent by not including it in the training of the non-parametric model and then repeating the process over all pixels in the study domain. In general, prominent 'mtry' is obtained by this method in extending the sample size, which prevents overfitting. Applying this validation method, the model has great skill on both the training dataset and the unseen test data. To strengthen the validation results, we also presented a cross-validation using leave-one-year-out analysis. Namely, each year of the database was hold out for validation, and the model was trained on the rest of the years (ten years). The model validation results based on the leave-one-pixel-out and leave-one-year-out are described in new section 4.2.

The temporal coverage of the data is mentioned in section 3.1. The temporal resolution of all precipitation products used in this study is three hourly.

2) There is also no information on avoiding overfitting. One of the challenges in data driven methods is overfitting (i.e. the method is so fine tuned to the training data, and has larger errors when applied to new datasets). I don't see any discussion of this in the paper. For example how did you choose to use 1000 trees in the model? Are there noticeable differences between the performance of the method during training and validation?

**Ans:**
In section 3.1 of revised manuscript we explain the overfitting issue. Quantile Regression Forests (QRF) uses bagged version of decision trees and obtain a lower test error by variance reduction (Meinshausen, 2006). Higher number of trees reduces the variance of the model. So, increasing the number of trees in the ensemble won't have any impact on the bias of the model. Furthermore, a higher variance reduction can be achieved by decreasing the correlation between trees in the ensemble. Therefore, QRF utilizes the optimal number 'mtry' (size of the random subset of predictors) for split point selection at each node. It will introduce some randomness in to the ensemble to reduce the correlation between trees, which helps to avoid overfitting (Meinshausen, 2006). In general, prominent 'mtry' is obtained by cross validation methods in extending the sample size. The application of the machine learning tools can manipulate the training data in such a way that the actual results expected from the unseen data can be quite different from the evaluated results using the training data set, which is called overfitting. Therefore, in this analysis, we used leave-one-out validation, discussed above, to prevent

overfitting and produce reliable results. Applying this validation technique, the model has good skill on both the training dataset and the unseen test data.

As an additional evaluation step, we also applied the leave-one-year-out cross validation method to evaluate the model's predictive accuracy over a longer unseen data record, which is discussed in section 4.2. The performances of the combined product, based on the leave-one-year-out cross validation are presented in Figures 10 and 11 of revised manuscript. Results for NCRMSE are shown in Figure 10, which are consistent in terms of the reduction of the random error for both seasons (warm and cold) as well as precipitation percentile ranges. The random error decreases with increasing scale and for all cases, results from combined product are associated with an error reduction (relative to other products) in the order of 52%-98%. Overall, results indicate that the random error of the combined product is significantly lower than those of the individual precipitation products used in this study.

The performance of the estimates for the model was also evaluated in terms of systematic error, as shown in Figures 11. Results show that the magnitude of systematic error for combined product is substantially lower than for individual precipitation products. We found that the magnitude of BR values (0.5-0.8 for moderate to high rain rates) was close to estimates of the model. Overall, BR values are closer to 1 for moderate to high rain rates in both seasons for the combined product than the other individual datasets, which indicates that QRF is able to reduce the systematic error in moderate to high rain rates.

The performance of the combined product, based on the one-year-leave-out cross validation, is found to be very similar to the results (Figures 6 and 7) determined based on the leave-one-pixel-out cross validation. Both validation approaches demonstrated that our model is able to reduce both systematic and random error components, which indicates that our model was appropriately trained and does not exhibit limitations due to overfitting.

In section 3.1 of the revised manuscript we elaborately discussed the reason of choosing 1000 trees in the model. As we discussed, higher number of trees would reduce the variance of the model and help to avoid overfitting. Therefore, the size of the forest should be relatively large for the stabilizing effect of many trees. For the Quantile Regression Forests, trees are grown as in the standard random forests algorithm and bagged versions of the training data are used for each of the k = 1000 trees to determine the optimal number mtry (Meinshausen, 2006). Therefore, to demonstrate the stability of QRF, the default value (k=1000) is chosen throughout all simulations.

3) How are the ensembles generated? No information is provided on how each ensemble member is initialized and generated using the QRF trained on the data.

**Ans:**
This aspect is elaborately explained in section 3.1. For QRF model, we initialize a random forest of 1000 trees for each terminal node of each of the classified dataset. We calculate 95% prediction intervals for each grid. QRF utilizes the same weights to calculate the empirical distribution function. When, X is predictor variable and Y is response variable, QRF utilizes a

weighted average of all trees for the predicted expected response values to calculate the empirical distribution. Subsequently, to conduct the hydrological simulations in this study, we resampled from the empirical distribution function 20 times per grid cell to obtain "reference"-like rainfall ensemble members.

**4) The results provided in section 4 needs to be clarified whether they are based on the data used in training or the data used in validation, or a mixture of both.**

**Ans:**
Section 4 of revised manuscript has been updated based on the results from the two different validation scenarios (leave-one-pixel-out and leave-one-year-out) where we explain the split of the dataset in training and unseen test. Our study period spans eleven years (2000–2010) and we validated all 11 years belonging in our dataset through the iterative cross leave one year out validation. The leave-one-pixel-out leaves one pixel (for the entire 11 years) at the time out and iterates over all pixel in the study domain. These two validation processes are substantiated in section 4.

**5) The low value of NCRMSE for the small basins report in Page 11, Line 6 is a signal of overfitting in the algorithm. This is another indication that overfitting should be analyzed in depth.**

**Ans:**
In the revised manuscript, detailed discussions on these aspects are provided in section 4.2. We hope that the updated analysis/results demonstrate that overfitting is not an issue.

**6) Page 6, Lines 10-18: Please clarify if different trees are developed for the three groups that you introduce at the beginning of the paragraph. You have introduced four groups at the end (warm-high, warm-low, cold-high and cold-low) but there is no reference to the categorization of products based on their rain detection (group 1-3 in lines 11-12).**

**Ans:**
Yes, we developed different trees for the three groups. If we grow similar kind of trees, every sampling will be equal, which affects the model performance. As we mentioned, (QRF) uses bagged version (bootstrapped aggregating) of decision trees by randomly sampling from bootstrapped sample which reduces variance and helps to avoid overfitting to improve the stability and accuracy of our proposed machine learning algorithms. That is the whole idea of choosing ensemble method where trees grow independently because of the combination of bootstrap samples and random drawing of variables.

Specifically, we classified available rainfall estimates from all the products (three satellite and reanalysis) into three satellite detection scenarios: (1) all rainfall products report rainfall greater than zero (2) all rainfall products report zero rainfall; and (3) at least one product reports nonzero rainfall. Then, for each scenario, we created 4 groups: warm period-high elevation, warm period-low elevation, cold period-high elevation, and cold period-low elevation to develop 12 QRF combination models. We created these categorizations to keep similar types of dataset together. If we mix data, our model will not be efficient (results are not presented in the paper).

Also, we would like to note that the QRF model is expected not to capture well very low and extremely high values due to the weakness of the empirical distribution function to model probabilities close to 0 or 1. The distribution of proper sample size plays an important role in empirical distribution function. Therefore, very large sample sizes required for low and extremely high values to quantify the rate of convergence to the underlying cumulative distribution function. This is the reason we categorized our dataset from above mentioned procedure.

-Minor Comments:
1) Why did you choose to use PERSIANN product instead of the newer version PERSIAN-CCS?

**Ans:**
In this study we chose to use gauge adjusted satellite precipitation products: 3B42 (V7), CMORPH and PERSIANN. The gauge-adjusted PERSIAN-CCS is not available over the Iberian Peninsula. Using the PERSIAN-CCS in precipitation error analysis is a good suggestion that we could investigated in a future research.

2) In section 2.3, please include details on how you have downscaled the 0.5 degree reanalysis product to 0.25 degree to be consistent with other products.

**Ans:**
This is clarified in the revised manuscript (see section 3.1)

3) In section 2.4, please include the version number of the ESA-CCI product.

**Ans:**
It is included in section 2.4.

4) Page 8, Line 2: What does actual uncertainty mean? Do you mean uncertainty in the reference product? If so, please explain how a UR=1 will provide the best estimate of the uncertainty in the reference product

**Ans:**
This issue is clarified in the revised manuscript. To achieve accurate and successful prediction, comparatively small prediction intervals are expected. The UR value 1 means best estimate of the actual uncertainty which indicates the maximum possible uncertainty of the prediction interval. Uncertainty Ratio (UR) quantifies the prediction interval width relative to the magnitude of observations. UR value close to 1, indicates confidence intervals being in the order of magnitude of the predicted values.

**Response to Short Comment #1 (SC1):**

I quickly read the paper by Md Abul Ehsan Bhuiyan et al. as I am very interested to the proposed methodology. Indeed, as the authors might know, we are working on the combination of state-of-the-art precipitation products (e.g., CMORPH, PERSIANN, 3B42) and satellite soil moisture data (e.g., ESA CCI SM) for improving satellite rainfall

I believe the paper is well written and clear. The final results are very encouraging. However, in my opinion a better description of the different steps involved in the procedure should be given. I reported below my comments/suggestions that I guess could be used from the authors for improving the paper's relevance.

1) As mentioned above, I am very interested to understand the contribution of the different datasets to the final combined precipitation dataset. What is the contribution of the satellite products with respect to the reanalysis? Which is the contribution of satellite soil moisture data? And of air temperature? I believe that running the QRF model in different scenarios considering different subsets of data will easily allow replying to these questions.

**Ans:**
We included a new section in revised manuscript (section 4.1), which elaborately explains the contribution of the different datasets to produce the final combined precipitation dataset. We have also updated analysis to demonstrate the relative importance of each variable. However, quantifying the relative change in performance for different scenarios, although very interesting, is not within the scope of this work. We will consider it in a future investigation.

2) Actually, if I well understood, the same data period is used for the calibration and the assessment of the combined precipitation dataset. It is not fair in the comparison with the single products. Likely, a split of the data in a calibration/validation period is needed.

**Ans:**
We explain about this matter in the revised manuscript (section 3.1). We also included one new section to strengthen the validation results, which are presented in section 4.2, where we extensively validated the technique through a leave-one-pixel-out and leave-one-year-out cross-validation analysis.

3) What is the final objective of the paper? If the authors want to provide a superior rainfall dataset, it should be tested against the SAFRAN reference dataset. What are the differences in the performance of hydrological modelling between SAFRAN and the combined dataset? This analysis might provide interesting insights.

**Ans:**
The main objective of this work is to present a method for optimally blending global precipitation dataset. We are presenting a blending technique that leads to an improved characterization of precipitation estimation uncertainty through an optimal combination of precipitation and other datasets, but we do not claim that we have examined exhaustively the combinations of variables or products that can potentially lead to a "superior product". In fact we hope that this work will trigger the interest of the community (as in your case) to investigate these aspects in more detail.

Evaluation metrics for rainfall and streamflow simulations are reported in the manuscript based on SAFRAN and SAFRAN-forced simulations as reference respectively. Please see Figs 6, 7, 10, 11, 12, and 13 and corresponding text in manuscript.

4) (MINOR) Among the different satellite rainfall products, PERSIANN and CMORPH should be the versions only based on satellite data. Differently, 3B42 (V7) is corrected with rain gauge observations. Therefore, the comparison between them is not fair, and I suggest in using the real-time version of TMPA (3B42RT) for a more interesting comparison.

**Ans:**
All satellite products used correspond to the gauge adjusted versions.